# Effects of High-Load Bench Press Training with Different Blood Flow Restriction Pressurization Strategies on the Degree of Muscle Activation in the Upper Limbs of Bodybuilders

**DOI:** 10.3390/s24020605

**Published:** 2024-01-17

**Authors:** Kexin He, Yao Sun, Shuang Xiao, Xiuli Zhang, Zhihao Du, Yanping Zhang

**Affiliations:** 1School of P.E. and Sports, Beijing Normal University, Beijing 100875, China; 11112018073@bnu.edu.cn (K.H.); 18074334809@163.com (S.X.); 2College of Physical Education, China University of Mining and Technology, Xuzhou 221116, China; sunyao5188@163.com; 3College of Physical Education (Main Campus), Zhengzhou University, Zhengzhou 450052, China; zxl_zzu_edu@126.com; 4School of Sports Science, Jishou University, Jishou 416000, China

**Keywords:** blood flow restriction, bench press, upper-extremity muscle groups, muscle activation level

## Abstract

**Background**: The aim of this study was to investigate the effects of different pressurization modes during high-load bench press training on muscle activation and subjective fatigue in bodybuilders. **Methods**: Ten bodybuilders participated in a randomized, self-controlled crossover experimental design, performing bench press training under three different pressurization modes: T1 (low pressure, high resistance), T2 (high pressure, high resistance), and C (non-pressurized conventional). Surface EMG signals were recorded from the pectoralis major, deltoid, and triceps muscles using a Delsys Trigno wireless surface EMG during bench presses. Subjective fatigue was assessed immediately after the training session. **Results**: (1) Pectoralis major muscle: The muscle activation degree of the T1 group was significantly higher than that of the blank control group during the bench press (*p* < 0.05). The muscle activation degree of the T2 group was significantly higher than that of the C group during the bench press (*p* < 0.05). In addition, the muscle activation degree of the T2 group was significantly higher than that of the T1 group during the first group bench press (*p* < 0.05). (2) Deltoid muscle: The muscle activation degree of the T2 group during the third group bench press was significantly lower than the index values of the first two groups (*p* < 0.05). The muscle activation degree in the experimental group was significantly higher than that in the C group (*p* < 0.05). The degree of muscle activation in the T2 group was significantly higher than that in the T1 group during the first bench press (*p* < 0.05). (3) Triceps: The muscle activation degree of the T1 group was significantly higher than the index value of the third group during the second group bench press (*p* < 0.05), while the muscle activation degree of the T2 group was significantly lower than the index value of the first two groups during the third group bench press (*p* < 0.05). The degree of muscle activation in all experimental groups was significantly higher than that in group C (*p* < 0.05). (5) RPE index values in all groups were significantly increased (*p* < 0.05). The RPE value of the T1 group was significantly higher than that of the C group after bench press (*p* < 0.05). The RPE value of the T1 group was significantly higher than that of the C group after bench press (*p* < 0.05). In the third group, the RPE value of the T1 group was significantly higher than that of the C and T2 groups (*p* = 0.002) (*p* < 0.05). **Conclusions**: The activation of the pectoralis major, triceps brachii, and deltoid muscles is significantly increased by high-intensity bench press training with either continuous or intermittent pressurization. However, continuous pressurization results in a higher level of perceived fatigue. The training mode involving high pressure and high resistance without pressurization during sets but with 180 mmHg occlusion pressure and pressurization during rest intervals yields the most pronounced overall effect on muscle activation.

## 1. Introduction

BFRT, also known as blood flow restriction training, is an exercise training method that uses a binding cuff at the near end of the limbs of the participants to completely block venous blood return, while reducing but maintaining arterial blood flow in the muscles, resulting in blood accumulation in the limbs, and triggering a stronger stress physiological response than traditional training [1]. BFRT is widely used in competitive sports training, mass fitness, and sports injury rehabilitation [2].

At present, international experimental studies on blood flow restriction show that this training method is a mainstream research topic, and the research consensus is that low-intensity resistance training combined with different pressure interventions can be superior to non-pressure training in strength performance. When pressure reaches a certain threshold, low-intensity blood flow restriction can cause strength gains and muscle hypertrophy similar to traditional high-intensity resistance training. Domestic and international exercise experiments on pressurized resistance have resulted in a series of studies on the effects of different pressurized resistance modes on muscular strength and endurance. In these studies, common compression resistance training mode combinations include low-intensity exercise (20%1 RM~30%1 RM) combined with low, medium, and high occlusion pressures (100 mmHg~300 mmHg) [3], moderate-intensity exercise (30%1 RM to 50%1 RM) combined with low-to-medium occlusion pressures (100 mmHg to 200 mmHg) [4], and high-intensity exercise (70%1 RM) combined with low occlusion pressure (100 mmHg~150 mmHg) [5,6,7]. However, little research has been conducted on the pressurized resistance mode of high-intensity exercise combined with medium and high pressures, mainly because sustained high occlusion pressures can lead to a significant increase in the risk of potential safety issues such as cardiovascular injuries, muscle injuries, and other injury risks [8], which to some extent can lead to the occurrence of sports injuries.

From the current stage of experimental studies on pressurized resistance at home and abroad, the means of pressurization include continuous pressurization and intermittent pressurization. Continuous pressurization consists of maintaining continuous pressurization throughout the entire exercise period, including rest between sets, while intermittent pressurization consists of two modes: pressurization during the exercise period but depressurization during the intermittent period or depressurization during the exercise period and pressurization during the intermittent period. Two recent experimental studies [9,10] were conducted by combining moderate-to-high intensity exercise with a high degree of occlusion pressure by arranging the participants to have no pressurization during exercise and medium-to-high pressurization during the rest intervals between sets and conducting an experimental study on acute resistance training of the lower extremities. This kind of pressurization intervention can prevent the safety problems caused by the continuous restriction of blood flow due to the continuous high pressure as well as address the paucity of research regarding a high-pressure high-resistance model in pressurized resistance training. It should be noted that the above two experimental studies focused on the effects of a high-pressure high-resistance training model of pressurized resistance training on the muscle strength and muscle dimensions of the lower extremities, but the neuromuscular properties of this model have not been examined in the upper-extremity muscles. Previous studies have clearly indicated that pressurized resistance training can bring about greater muscle activation than traditional resistance training and that pressurized stimulation has a better effect on neuromuscular properties [11,12]. Therefore, it is reasonable to investigate the acute changes in muscle activation in the upper extremities during high-pressure high-resistance bench press training by means of changing the mode of pressurization.

## 2. Materials and Methods

### 2.1. Participants

This study calculated the number of participants through G*Power software 3.1 version, and finally selected 10 participants. The participants in this experiment were 10 bodybuilders from the College of Physical Education and Sports, Beijing Normal University, and the inclusion criteria for the participants were as follows: (1) more than three years of resistance training and proficiency in the bench press exercise; (2) a 1 RM in a bench press of at least 1.2 times their body mass; and (3) no athletic injuries in the last six months. The basic information of the participants is shown in Table 1. The purpose of this experiment, the method of implementation, and the possible associated risks were fully explained to the participants before the experiment, and consent was obtained from the faculty coach and the athletes. Before the test, the participants were informed of the training movements involved in the experiment and the interventions that differed from normal training, and each subject was allowed to wear the pressurization equipment during training for familiarization one week before the experiment. All participants signed informed consent forms prior to the experiment, which strictly followed the Helsinki Declaration [13] and passed the review of the Ethics Committee of Zhengzhou University (ZZUIRB2022-JCYXY0016). The experimental site was the National Fitness Center of Physical Education College of Beijing Normal University.

Inclusion criteria: (1) bodybuilders aged 20–25 years; (2) no major diseases or chronic diseases; (3) proficient in standard bench press training movements; (4) bench press 1 RM level of at least 1.2 times body mass.

Exclusion criteria: (1) muscle strain, tendon inflammation, fracture, and other sports injuries that occurred in the past six months; (2) rich experience in resistance training; the total training years must reach three years or more; (3) use of specialized illegal drugs that promote muscle growth; (4) Reliance on protective equipment such as power belts, wrist guards, and other equipment to complete the lying push.

### 2.2. Methods

#### 2.2.1. Experimental Design and Intervention Program

The 72 h before the formal experiment was the preparation stage. First, the participants were recruited, and then the basic information of the participants was recorded. The parameters included age, height, body mass, and bench press 1 RM for each subject. See Figure 1 for details

On the day of the formal experimental test, participants first warmed up and then performed a maximum voluntary isometric contraction (MVIC) test and pressurized bench press surface electromyography test on the target muscles of the upper limb. Each subject performed three different modes of bench press training: C (no-pressurization mode): bench press training at an exercise intensity of 70%1 RM; T1 (low-pressurization, high-resistance mode): bench press training at an exercise intensity of 70%1 RM with continuous pressurization and an occlusion pressure of 100 mmHg; and T2 (high pressurization, high resistance): bench press training at an exercise intensity of 70%1 RM with intermittent pressurization denoted by a depressurization phase during the exercise period and a pressurization phase with an occlusion pressure of 180 mmHg during the intervals between the sets. The participants performed all three modes of training protocols as 3 continuous sets of 8 repetitions each, with a 2 min interset rest interval for bench press training. In this study, the right target muscle groups of the pectoralis major, deltoid, and triceps brachii were selected for the MVIC test (the order of muscle testing was randomized and parallel), and the changes in electromyography were recorded by using a 3-channel Delsys Trigno wireless surface electromyography signal acquisition system at the same time as the MVIC test. Five minutes after the completion of the MVIC test, the participants underwent pressurized bench press training, and Theratools BFR pressurized bench press equipment was used. Theratools BFR pressurization equipment was used for pressurized intervention under the following conditions: the bundle pressure was 30 mmHg; the location of the pressurized cuff was near the proximal end of the arm; the surface EMG signal acquisition was synchronized with the pressurized bench press training; and the sequence of pressurized bench press conditions in different modes was randomized and counterbalanced. The time interval between the two different modes was 72 h, not only to prevent muscle damage caused by the pressurized resistance training from affecting the participants’ test status [14] but also to reduce the mutual interference effect between the different modes.

#### 2.2.2. Testing of Experimental Indicators

(1)Bench Press 1 RM Test

Seventy-two hours prior to the experimental phase, participants performed a 1 RM bench press test, and their personal data were collected, including 1. height, 2. body mass, 3. age, and 4. shoulder width. Participants were sequenced through a parallel and randomized lottery for the experimental protocol. After data collection was completed, a bench press 1 RM test was performed using a standardized warm-up procedure for each test using a bicycle ergometer with an upper-body component for approximately 5 min with a resistance of 100 W and a pedaling frequency between 70 and 80 rpm. Next, a warm-up was performed for upper-body muscle groups, including mobility exercises for the shoulders and chest. After the general warm-up was completed, a specialized warm-up was performed. Participants performed 15, 10, and 5 repetitions of the bench press using 20%, 40%, and 60% of the estimated 1 RM. After warming up, the 1 RM test is as follows: (1) increase body mass by 4–9 kg from 80% of the estimated 1 RM per successive attempt, conservatively repeat 3–5 times, then rest for 2 min. (2) Continue to increase the body mass by 4–9 kg, then complete 2–3 repetitions, rest, then increase by 4–9 kg while trying to lift the 1 RM. (3) If one attempt is successful, continue to increase the body mass by 4–9 kg, but if the attempt fails, reduce the body mass by 2–4 kg and measure 1 RM over the course of 3–5 attempts. This process was repeated until failure. The 1 RM for all participants was determined in 5 trials.

In the 1 RM test and the formal experiment, participants were asked to bench press with a constant rhythm and trajectory (using a metronome to control the subject’s rhythm, the duration of the centrifugal phase (point A → point B) was 2 s, and the centripetal phase (point B → point C) was 1 s to try to ensure consistency of the movement time). Participants were instructed to keep their head, shoulders, and hips in contact with the bench, lower the barbell until it touches the chest during its descent, and fully extend the elbows at the end of the concentric phase to achieve a valid repetition in the bench press movement.

Participants’ hands were placed on the barbell in a fixed position equal to 150% of the individual’s peak shoulder width. Participants were not allowed to use weightlifting belts, wrist wraps, elbow sleeves, or other assistive equipment during the exercise period, and all repetitions were directly supervised by an experienced physical trainer.

(2)Maximum random isometric contraction (MVIC) test for target muscle groups of the upper limb

On the day of the official test, after the same warm-up as in the bench press 1 RM test, the MVIC test was first performed using a 3-channel Delsys Trigno wireless surface EMG signal acquisition system to measure and analyze the surface EMG signals of the muscles. Based on the biomechanical characteristics of the bench press training maneuver, three muscles of the upper limb were included, pectoralis major, deltoid, and triceps brachii, and the electrodes were located on the muscle belly of each of these three muscles with reference to anatomical characteristics. Before placing the gel-coated self-adhesive electrodes, the hair covering the muscle was shaved, and the skin was cleaned of dirt and sweat with alcohol, which reduced the skin’s resistance while still being able to ensure effective attachment of the electrode sensors. According to Konrad’s program [15], the integrated EMG values were collected for each muscle under the MVIC test. The test method was as follows (2 MVIC tests were performed for each muscle):

① Pectoralis muscle (Pectorals): The elbow joint was at 90 degrees, push-ups were performed, the tester pressed down on the subject’s shoulders and gradually exerted force, the subject pushed upward as hard as he could to push off the ground, and the tester put up resistance to him and insisted on the action for 3–5 s to collect the electromyographic data of the pectoralis muscle. ② Anterior Deltoid: The subject was in a sitting position, the arm was flexed forward to the epigastric region, the tester pressed the subject’s wrist, the subject’s arm was flexed upward with maximum force, the tester exerted downward force against it, and the electromyographic data of the anterior deltoid were collected by maintaining the action for 3–5 s. ③ Triceps brachii: The subject was in a sitting position, the arm was naturally hanging down, the upper and lower arm were at 90 degrees, the tester grasped the wrist of the subject, the subject performed a lower-arm extension movement downward with maximum force, and the tester provided upward resistance, maintaining the movement for 3–5 s to collect the electromyographic data of the triceps brachii muscle.

(3)Pressurized bench press training surface EMG test

After completing the MVIC test, the participants took a rest for 5 min. During this interval, the tester implemented pressure intervention on the participants as a follow-up test. In this experiment, Theratools BFR pressure equipment was adopted. The equipment consists of a pressure pump and a binding cuff. The binding method is Velcro tape, and the pressurized part is the tropanes of the deltoid muscle of the athlete’s arm. The binding pressure is 30 mmHg, the blocking pressure during exercise is 100 mmHg at low pressure and 180 mmHg at high pressure, respectively, and the pressurized method is continuous pressure at low pressure; that is, pressure is applied throughout the bench press. The high pressure adopts intermittent pressure: intermittent pressure in each group of exercises, and pressure is reduced during exercise. The sequence of EMG testing on the surface of the bench press in different pressurization modes was randomized and counterbalanced, and the time interval between the two different pressurization modes was 72 h, with a 2 min interval between each set. The pressurized bench press surface EMG test was performed after all preparations were made. The maximum heart rate and immediate postexercise blood pressure were selected as indirect indicators to judge the training intensity and training safety of the different conditions for the participants, and the greater the maximum heart rate and immediate postexercise blood pressure were, the greater the intensity of the pressurized resistance exercise intervention was considered to be [3].

(4)Subjective fatigue (RPE) test

Prior to the start of the experiment, the standardized meanings of the RPE values were explained to the participants, and after each set of bench press training in the different pressurization modes, the participants were tested for subjective fatigue, and the grades were recorded. The test was performed using Zourdos’ novel scale [16], and the levels ranged from 1 to 10, in which the first four levels were scored using subjective exertion: levels 1–2 indicated no exertion at all, and levels 3–4 indicated slight exertion. Levels 5–10 were scored using the number of repetitions in reserve (RIR): levels 5–6 indicated that the subject could have performed 4–6 more repetitions at the conclusion of that pressurized bench press set, level 7 indicated that 3 more repetitions were possible, level 8 indicated that 2 more repetitions were possible, level 9 indicated that 1 more repetition was possible, and level 10 indicated exhaustion.

### 2.3. Statistical Methods

The EMG data analyzed above were analyzed using Excel 2010 and SPSS 17.0 statistical software, and all the data were expressed in the form of mean ± standard deviation (M ± SD). A repeated-measure two-way ANOVA (number of exercise conditions × mode of pressurization) was used to statistically analyze the RMS standard values of upper-limb muscle groups during high-intensity bench press training in different modes of pressurization, and LSD was used for multiple comparisons of means when the interaction was significant. Finally, subjective fatigue after training in each condition was statistically analyzed using paired-sample *t* tests. The significance level of the test of difference for all the above analyses was taken as *p* < 0.05.

## 3. Results

### 3.1. Multiple Comparison Analysis

A two-way (model of pressurization × exercise condition) repeated-measure ANOVA revealed that the mode of pressurization had a significant effect on the change in activation values of the pectoralis major, deltoid, and triceps brachii (*p* < 0.05); the exercise condition had a significant effect on the change in activation values of the deltoid and triceps brachii (*p* < 0.05); and there was no significant effect of the mode of pressurization and exercise condition on the change in activation values of all muscles (*p* > 0.05).The EMG raw signal is detailed in Figure 2.

### 3.2. Changes in %MVIC Values of Target Muscles in Each Set of Bench Press Training in Different Pressurization Modes

The results showed (Table 2 and Table 3) that pectoralis major muscle activation values were significantly higher in T1 than in C when performing the first (*p* = 0.027), second (*p* = 0.005), and third sets of bench presses (*p* = 0.011) (*p* < 0.05); muscle activation values were significantly higher in T2 than in C when performing the first (*p* = 0.000) and second (*p* = 0.003) sets of bench presses (*p* < 0.05); and muscle activation values were also significantly greater in T2 than in T1 when performing the first (*p* = 0.016) set of bench presses (*p* < 0.05).

Deltoid muscle activation in T2 was significantly lower than that in T1 (*p* = 0.032) and T2 (*p* = 0.000) during the third set of bench presses (*p* < 0.05); muscle activation in T1 was significantly higher than that in C during the first (*p* = 0.000), second (*p* = 0.014), and third (*p* = 0.015) sets of bench presses (*p* < 0.05); and muscle activation in T2 was significantly higher than that in C during the first (*p* < 0.000), second (*p* = 0.000), and third sets (*p* < 0.000) of bench presses (*p* < 0.05). The value of muscle activation in T2 was also significantly greater than that of T1 (*p* < 0.05) when performing the second set of bench presses (*p* = 0.000).

Triceps brachii muscle activation was significantly greater in T1 (*p* = 0.014) during the second set of bench presses and in T2 during the third set of bench presses (*p* < 0.05) but significantly less than in Condition I (*p* = 0.005) and T2 (*p* = 0.002) during the third set of bench presses (*p* = 0.000). Muscle activation values were significantly higher than those of C (*p* < 0.05), but during the third set of bench presses (*p* = 0.000), muscle activation was significantly lower than that of C (*p* < 0.005). T2 muscle activation values were significantly higher than those of C when performing the first (*p* = 0.000) and second (*p* = 0.000) sets of bench presses (*p* < 0.05), but the degree of muscle activation during the third set of bench presses (*p* = 0.000) was significantly less than that of C (*p* < 0.05); see Table 4 for details.

### 3.3. Subjective Fatigue Results after High-Load Bench Press Training in Different Pressurization Modes

The results showed that with the increase in the number of bench press sets (Table 5), both the pressurized experimental condition and the nonpressurized C RPE index value increased significantly (*p* < 0.05); after the first bench press set, the T1 RPE value was significantly greater than the index value of the C (*p* = 0.030); after the second bench press set, the T1 (*p* = 0.016) RPE value was significantly greater than the index value of the C; after the third bench press set, the T1 RPE value was significantly greater than the index values of C (*p* = 0.022) and T2 (*p* = 0.002) (*p* < 0.05), and there was no significant difference between T2 and C (*p* = 0.121) (*p* > 0.05).

## 4. Discussion

The main purpose of this study was to analyze the effects of high-load bench press training with different pressure modes on upper-limb muscle activation characteristics and the subjective fatigue of bodybuilders.

(1)Analysis of changes in the pectoralis major activation level

This study found that the degree of pectoral major muscle activation in bodybuilders during bench press training was significantly higher than that in group C. The results of this study support the study of Che Tongtong [12], who arranged 10 basketball players to perform four groups (30–15–15–15) of low-intensity pressure (160 mmHg) bench press training, and also observed that the activation degree of the pectoralis major muscle in the pressure group was significantly greater than that in the non-pressure group. A large number of previous studies have shown that during the contraction process of skeletal muscle with blood flow restriction [7,17,18,19,20,21,22], there will be intramuscular hypoxia due to the limitation of blood flow, as well as the accumulation of lactate, hormones, and other metabolites (overload). A series of stress responses through the stimulation of afferent nerve centers III and IV can be fed back and cause a large amount of muscle fiber recruitment, which may lead to the inhibition of α-motor neurons. Thus, the body may lead to inhibition of α-motor neurons so that the organism may recruit more muscle fibers to maintain force and prevent conduction failure, which is manifested by a significant increase in muscle activation. In addition, a recent review article [23] suggests that the application of pressurized stimuli during resistance training leads to an increase in the proportion of anaerobic metabolic energy supply, resulting in rapid fatigue of type-I muscle fibers, which induces an increase in the recruitment of high-threshold type-II muscle fibers, which further increases muscle activation.

Che et al. [12] found that in four groups of low-intensity pressure training, the activation degree of pectoralis major was significantly greater than that of the non-pressure group only during the bench press training of the third and fourth groups. However, the difference in this study is that the activation degree of pectoralis major in the three groups of bench press training was significantly greater in the pressure experimental group than in the C group, and the activation degree of pectoralis major in the first two groups of bench press was significantly higher in the T2 group than in the C group. However, there was no significant change between the third group of squats. The possible reasons can be attributed to differences in exercise intensity and specialization. First of all, previous studies only adopted 30%1 RM, which may be due to the low metabolic pressure in the first two groups and failed to induce the recruitment of muscle fibers with a higher threshold due to the low-intensity training in the first two groups. In this study, the exercise intensity of 70%1 RM was used for training, so a large mechanical load pressure was generated during the bench press in the first group, resulting in a gradual increase in metabolic pressure. Therefore, high-intensity resistance training can recruit high-threshold muscle fibers more quickly, so a significant increase in pectoralis major muscle activation was observed earlier in this study (in the first set of bench presses), and this study also confirmed that exercise intensity has a significant effect on neuromuscular adaptation induced by pressure training [24]. However, there was no significant change between the conditions during the third set of bench presses. The possible reasons can be attributed to the differences in exercise intensity and specialization: first, the previous study only used a 30%1 RM load, while the present study involved training with 70%1 RM as the exercise intensity, and the mechanical load was significantly greater than that of the former; therefore, different metabolic pressures were generated so that due to the lower metabolic pressures of low-intensity training in the first two sets, higher-threshold motor unit recruitment did not occur, whereas the high-intensity resistance training was able to more quickly recruit those motor units. Second, the participants in the present study were bodybuilders, whereas those in the previous study were basketball players. Theoretically, the bodybuilders underwent more resistance training and may have had greater neuromuscular recruitment of the pectoralis major muscle during the bench press. Relatively speaking, the daily training of bodybuilding athletes mainly focuses on resistance training. Bench press is also the core action of daily upper-limb training, and its frequency is significantly higher than that of other sports. Therefore, bodybuilders as participants in this bench press test showed a better performance of recruitment ability of the pectoral major muscle [25].

Notably, the present study showed that the degree of pectoralis major activation in T2 was significantly greater than that in T1 during the first set of bench presses, indicating that the use of a blood flow restriction pressurization strategy with pressure relief during the training period and high pressurization during the interset interval (180 mmHg) was more effective in promoting the degree of pectoralis major activation than continuous low pressurization (100 mmHg) during bench press training at an exercise intensity of 70%1 RM. In this regard, The above research results are consistent with the dose–effect relationship of pressure resistance training proposed by Lei et al. [23]; after reviewing the training effects of pressurized resistance training, Lei et al. [23] pointed out that there was an “inverted U-shaped” quantitative relationship between the amount of pressurization and muscle performance in pressurized resistance training, i.e., within the appropriate range of pressurization, with an increase in occlusion pressure, the stronger the occlusion effect, the greater the stimulation, and a greater stimulation will be more effective than continuous low pressure (100 mmHg). When the pressure reaches a certain high value, the stress and adaptation of the organism reach an optimal level, and the training effect is optimized; however, when the pressure is higher than the critical value, the metabolic load stimulus is too strong to exceed the limit of the organism’s adaptation, resulting in a decrease in the effect of training, which is related to the fatigue of the exercise. The results of the present study support the quantitative effect relationship proposed in the above study [23]. In addition, there was no significant difference between the two experimental conditions in the third set of bench presses, but the low-pressure T1 was significantly higher than C, but there was no significant difference in the high-pressure T2 condition. According to the above dose–effect relationship [23], it can also be inferred that the metabolic pressure caused by intermittent high-pressure training may exceed the limit of the participants, suggesting that this may be related to the gradual accumulation of greater exercise fatigue caused by high occlusion pressure.

(2)Analysis of changes in deltoid activation level

The results of the present study showed that deltoid activation was significantly higher in the three sets of bench press training in both the pressurized experimental conditions, T1 and T2, than in C. In addition, when the first set of bench presses was performed, T2 had significantly greater deltoid activation than T1. The results also support a previous study by the Che team [12], in which 10 basketball players performed five sets (30–15–15–15–15) of low-intensity stress (160 mmHg) bench press training and found that deltoid muscle activation was significantly greater under stress conditions than under non-stress conditions. However, Che’s study [12] found that deltoid muscle activation increased significantly only in the fourth group of bench press, while in this study, deltoid muscle activation was similar to or even superior to that of the pectoralis major muscle, including the third group of bench press under T2, and deltoid muscle activation under stressful T1 and T2 was significantly higher than that under stress-free C in the three groups of high-intensity training. The stronger effect of blood flow restriction on deltoid activation can be attributed to the site of the pressurized band. After conducting low-intensity pressure resistance training experiments, Loenneke et al. [26] pointed out that pressure stimulation during exercise training significantly reduced venous backflow, so that after subsequent compression band decompression, a pressure gradient conducive to peripheral blood flow into muscle fibers was quickly formed. Under the pressure gradient, extracellular fluid flooded into the cells and tissues of the exercising muscle group. At the same time, due to the stimulation of exercise training and stress, the accumulation of metabolic stress products will produce stronger stimulation of peripheral and central chemoreceptors. At this time, the metabolic stress products accumulated due to exercise training and pressurization stimulation will produce stronger stimulation to peripheral and central chemoreceptors, which will ultimately promote the synthesis of hormone proteins for myocyte growth and development. This may be the main reason for the more pronounced changes in muscle activation in the deltoid than in the pectoralis major muscle [27].

One thing that differentiates the change in the activation level of the pectoralis major is that the activation level of the deltoid muscle during the third set of bench presses performed during T2 was significantly less than the activation level during the first two sets of bench presses. This may be because as exercise continues, the intermittent high-pressure mode occurs due to increased metabolic stress and accumulation of metabolic waste products as a result of the greater occlusion pressure, which causes exercise fatigue and ultimately reduces the level of muscle activation in the participants.

(3)Analysis of changes in triceps activation

The results of this study showed that the triceps activation degree of the T1 group was significantly lower than that of the second group during the third set of bench presses, and the triceps activation degree of the T2 group was significantly lower than that of the first two groups during the third set of bench press; that is, with the progress of each group of bench press, the triceps activation degree of the pressure group was continuously reduced, while the reduction degree of the T2 group was greater than T1. The reason for this phenomenon may be that, on the one hand, the binding part of the upper-limb compression training is the upper arm; that is, the whole muscle of the triceps is bound by the compression band. In the case of continuous compression, although the blocking pressure is small (100 mmHg), the muscle will produce certain discomfort during the bench press with the progress of training, and the training effect will be reduced. On the other hand, pressure training requires participants to exercise under the dual stimulation of “pressure” and “resistance”. Therefore, with the continuous progress of exercise, this training mode leads to factors such as muscle and nerve disorder, continuous consumption of energy substances, and a decrease in energy metabolism and energy supply rate, resulting in the exercise-induced fatigue of participants and ultimately reducing the activation degree of triceps. It should be mentioned that according to the overall change in muscle activation degree, the third group of bench press training will lead to a greater decrease in the activation degree of the triceps muscle than that of the pectoralis major and deltoid muscles. The possible reason may be that the compression band binding part of the upper arm near the proximal heart during press bench press training basically covers the triceps muscle. The occluded pressure and metabolic pressure on the triceps reach the peak, so fatigue occurs most significantly.

The results of this experiment also showed that even though blood flow restriction resulted in a significant decrease in triceps activation as the bench press continued, the activation of the triceps was still significantly higher with blood flow restriction compared to nonpressurized traditional resistance training, and the changes in this muscle were basically the same as those in the pectoralis major and deltoid muscles, which supports the results of a previous study [12]. The results of the present experiment confirm that blood flow restriction is able to promote better neuromuscular properties of the triceps brachii muscle than no blood flow restriction.

### 4.1. Analysis of Subjective Fatigue Test Results

The research [11] confirms that RPE is a method to monitor subjective fatigue during exercise and to effectively monitor and regulate the intensity of loads in sports. The results of this study showed that RPE values increased significantly in all conditions as each set of bench presses was performed. It is not difficult to understand that, with or without pressurization intervention, continuous resistance training produces a certain amount of metabolic stress, which results in the body experiencing neuromuscular disruption, depletion of energy substances, a decrease in energy metabolism, an increase in the metabolic stress response, and an accumulation of metabolic wastes, which can result in performance decrements as well as physiological fatigue of the body; therefore, the RPE values of the experimental and C groups are increasingly large.

The present study also found that the T1 had significantly higher RPE values than the C in all three bench press sets, and the findings support previous research from Vieiea’s research team [28]. The design of a comparative test revealed that blood flow restriction produced a stronger subjective feeling of fatigue compared to traditional high-intensity resistance training. Schwiete et al. [29] also indicated that perceived pain was significantly higher in the pressurized resistance training experimental condition than in the nonpressurized C. Domestic scholars, such as Che Tong et al. [11] also observed that the RPE value of the blood flow restriction condition was significantly higher than that of the nonpressurized condition after putting basketball players through low-impact compression squats. However, in the third set, the T1 index value after bench press was significantly higher than that of the T2 condition, while there was no significant difference between T2 and C. This suggests that it may be more difficult for participants to use the continuous low-pressure mode of pressurization. Despite the large amount of pressurization, the intermittent high-pressure mode of pressurization with pressure relief during the exercise period can give participants a certain amount of time to rest during the pressure relief period, which can provide adequate time to restore the energy supply and remove metabolic waste so that subjective fatigue is lower. Therefore, the subjective fatigue of the participants was lower when using the intermittent pressurization mode, and this method is more favorable for bodybuilders to perform pressurized high-resistance training.

### 4.2. Practical Application

The purpose of this study is to compare the changes in the muscle activation degree of muscle groups in high-intensity bench press training under different pressure modes, explore the adaptability of the high-pressure high-resistance training model, and expand the application field of compression resistance training so as to provide a theoretical basis for bodybuilders to enhance neuromuscular adaptation and improve muscle mass (muscle hypertrophy/muscle recruitment).

### 4.3. Advantages of the Study

Intermittent pressure intervention was used in this study, and surface-integrated myography was used as an outcome index in high-intensity resistance training. This method can not only prevent safety problems such as cardiovascular injury and muscle injury due to the blood flow limitation caused by continuous high pressure but also fill the research gap in neuromuscular adaptation of a high-pressure and high-resistance training model.

### 4.4. Study Limitations

First, this study only monitored participants’ heart rate and blood pressure to control load intensity to prevent exercise risk. However, the implementation of pressure intervention in high resistance training also involves risk factors such as blood pressure, so hemodynamic indicators need to be monitored to improve the applicability of different pressure resistance training models. Secondly, this study mainly used surface-integrated myography and RPE as the main outcome indicators, but based on the characteristics of pressure training, changes in blood composition are also another important indicator. Since pressure training tends to lead to more metabolic stress, changes in the expression of metabolic stress products such as lactic acid, nitric oxide, and interleukin in blood components have not been thoroughly studied.

### 4.5. Future Research Directions

Further research could explore the persistence and potential long-term adaptation of the acute effects found in this study by conducting long-term compression resistance exercise interventions. Long-term trials will be designed to more fully assess the potential effects of exercise training on physical function, physiological parameters, and athletic performance. In addition, long-term experiments can provide more time windows to observe and analyze possible individual differences and adaptive changes, focusing on the design and execution of long-term experiments to gain insight into the long-term effects of exercise training and its potential mechanisms. It also provides a more reliable scientific basis for improving training programs and optimizing sports performance.

## 5. Conclusions

(1) In high-intensity pressurized bench press training, both continuous and intermittent pressurization significantly increased the activation of the pectoralis major, triceps brachii, and deltoid muscles, but the continuous pressurization method caused a stronger subjective fatigue sensation. (2) In terms of the overall training effect, the high-pressure, high-resistance training mode of no pressurization during the training period and pressurization during the intervals between sets with an occlusion pressure of 180 mmHg had the best enhancement effect.

## Figures and Tables

**Figure 1 sensors-24-00605-f001:**
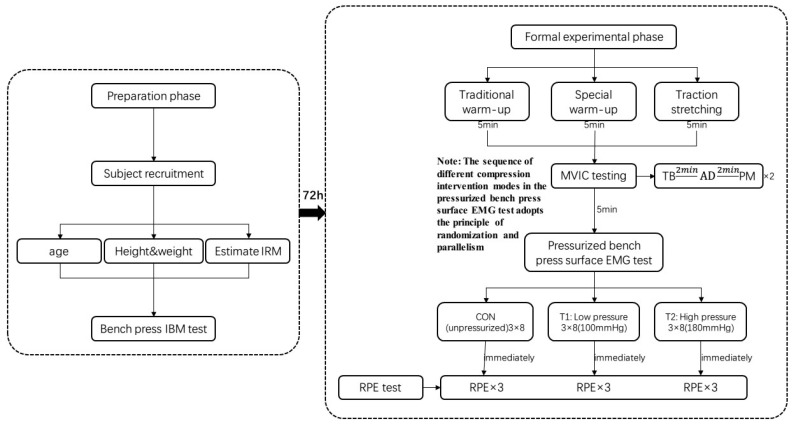
Flowchart of the experiment.

**Figure 2 sensors-24-00605-f002:**
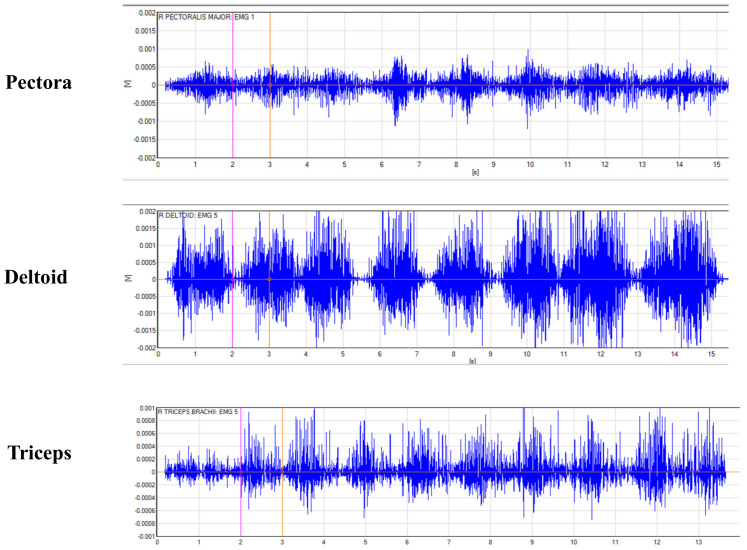
Raw EMG signals of the target muscle during bench press performed by one subject.

**Table 1 sensors-24-00605-t001:** Basic information of the participants.

Age	Height (cm)	Body Mass (kg)	Shoulder Width (cm)	Bench Press 1 RM (kg)
23.67± 1.73	174.22± 4.06	79.17 ± 8.28	44.50 ± 4.28	106.33 ± 10.48

**Table 2 sensors-24-00605-t002:** Results of Levene’s homogeneity test.

Pressurized Mode	Pectoralis Major Muscle (Across the Top of the Chest)		Deltoid Muscle (Over the Shoulder)		Triceps Brachii (Back of the Upper Arm)	
	F	P	F	P	F	P
pectoralis major muscle (across the top of the chest)	0.077	0.926	0.177	0.839	0.768	0.475
deltoid muscle (over the shoulder)	0.631	0.541	0.049	0.952	0.391	0.68
triceps brachii (back of the upper arm)	0.036	0.964	0.021	0.979	0.149	0.862

Based on the results of Levene’s homogeneity test, it was found that *p* > 0.05 for all groups, i.e., the sample variance was equal for all groups, thus allowing for a two-factor repeated-measures ANOVA.

**Table 3 sensors-24-00605-t003:** Effect of pressurization mode, exercise condition, and their interaction on %MVIC values of target muscles.

	Pectoralis Major Muscle (Across the Top of the Chest)		Deltoid Muscle (Over the Shoulder)		Triceps Brachii (Back of the Upper Arm)	
	F	P	F	P	F	P
Pressurized mode	15.931	0.000 *	57.209	0.000 *	41.198	0.000 *
sports condition	0.881	0.421	3.648	0.046 *	3.251	0.047 *
interaction	1.971	0.134	1.511	0.214	0.722	0.551

Note: “*” represents a significant difference; significant difference is *p* < 0.05.

**Table 4 sensors-24-00605-t004:** List of changes in %MVIC values for each muscle in each set of bench press training in different pressurization modes.

Muscle Name	Pressurized Mode	3 Sets of Bench Press Training (8 + 8 + 8)		
		Set I	Set II	Set III
pectoralis major muscle (across the top of the chest)	Sustained low pressure (T1)	43.73 ± 19.97 *Δ	46.30 ± 20.01 *	47.33 ± 21.05 *
	Intermittent high pressure (T2)	61.21 ± 23.27 *	53.46 ± 15.56 *	44.79 ± 18.61
	Unpressurized (C)	34.33 ± 16.64	34.15 ± 19.98	36.28 ± 19.19
deltoid muscle (over the shoulder)	Sustained low pressure (T1)	64.02 ± 15.09 *	58.39 ± 13.66 *#	58.56 ± 12.27 *
	Intermittent high pressure (T2)	73.85 ± 16.31 *§	75.40 ± 15.89 *§	65.85 ± 14.99 *
	Unpressurized (C)	48.83 ± 12.56	48.63 ± 12.88	48.85 ± 15.08
triceps brachii (back of the upper arm)	Sustained low pressure (T1)	46.42 ± 22.25 *	56.85 ± 16.45 *§	32.27 ± 19.75 *
	Intermittent high pressure (T2)	54.64 ± 13.79 *§	57.62 ± 13.46 *§	33.41 ± 16.79 *
	Unpressurized (C)	40.32 ± 17.43	47.46 ± 12.98	36.49 ± 18.37

Note: “*” indicates that there is a significant difference in the change in muscle activation level between the pressurized experimental condition and Condition C; “Δ” indicates that there is a significant difference in the change in muscle activation level between Condition T2 and Condition T1 in the first set of bench presses; “#” indicates that there is a significant difference in the change in muscle activation values between Condition T2 and Condition T1 in the second set of bench presses; “§” indicates that there is a significant difference in the change in muscle activation level values between the three sets of bench presses within each experimental condition, with a significant difference of *p* < 0.05.

**Table 5 sensors-24-00605-t005:** List of subjective fatigue RPE scores of participants after bench press training in different pressurization modes (*n* = 10).

Subjective Fatigue	Pressurized Mode	Training Group		
		Set I	Set II	Set III
RPE value	Sustained low pressure (T1)	6.22 ± 0.67 *§	7.11 ± 1.17 *§#	8.67 ± 1.22 *
	Intermittent high pressure (T2)	5.89 ± 1.05 §	6.80 ± 1.12 §#	8.00 ± 1.41 Δ
	Unpressurized (C)	5.00 ± 1.00 §	5.89 ± 0.60 §#	7.11 ± 1.17

Note: “*” indicates that there is a significant difference in the RPE value comparing the pressurized experimental condition with Condition C; “Δ” indicates that there is a significant difference in the RPE value between Conditions T1 and T2; “§” indicates that there is a significant difference in the RPE value comparing the first set of bench presses with the second and third sets within each condition. Within each condition, there were significant differences in RPE values between the first set of bench presses and the second and third sets; “#” indicates that there were significant differences in RPE values between the second and third sets of bench presses within each condition (*p* < 0.05).

## Data Availability

All data are published on the figshare platform: 10.6084/m9 figshare. 24412597.

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
