# Peer review of "Effects of High-Load Bench Press Training with Different Blood Flow Restriction Pressurization Strategies on the Degree of Muscle Activation in the Upper Limbs of Bodybuilders"

_sensors, 2024, doi:10.3390/s24020605_

Round 1
Reviewer 1 Report
Comments and Suggestions for Authors
General Comments
This study is an interesting approach to the blood flow restriction procedure. A major handicap to reading the material is the use of very long sentences throughout. This makes it difficult to understand the concept being presented. Also, multiple sentences throughout are separated by semi-colons (;) which also hampered the flow of reading. It seems reasonable to express these as independent sentences for easier reading.
Specific Comments
Line Comment
51-58 This is an example of a long sentence that is a bit hard follow for meaning. It could be turned into 3 sentences that might be easier to follow.
58-65 This sentence could be split at “….strength and endurance. In these studies, the common combination of pressurized resistance training modes included …..
159-165 This is another series of statements separated by semi-colons. You might consider separate sentences for easier reading.
216-221 Another example of a long sentence which could be divided.
223-226 This too is difficult to follow due to the stringing of multiple ideas in one sentence.
251 “The EMG data were statistically analyzed………..”
282-286 Consider separating these three or four analyses into different sentences for easier reading.
310-316 Again, consider separating these semi-colon phrases into different sentences for easier reading.
334 Should this be “In that study…” since you are referring to Cha’s basketball study?
338 Does [7, 16-21] belong here since it appears you are referring the Cha’s work?
354 Would it be easier to read this if you started a new sentence here with “In the present study, ….”?
387 To what study does “in the above study” refer; is it Lei et al.? Should you give a citation number to clarify?
391 At what time does your phase “at this time” refer?
399 Should “Che” be Cha?
402 Should “Che” be Cha and should you provide a reference number for it?
405 Does the phrase “even better than that of the pectoralis major” belong here since you mentioned the pectoralis major before that?
402-409 Again, this is a very long sentence that is hard to follow.
410-417 This is another long sentence that is difficult to read and comprehend the meaning.
431-454 More long sentence’s that would read better if they were divided.
438 Does “On the one hand” belong here since it was stated earlier?
453 What does “the big arm” mean?
465 Does the reference [11] belong here?
481 Should “Che” be Cha?
483 “….. having basketball players perform….”
Comments on the Quality of English Language
The main issue seems to be that of lengthy sentences that are difficult to follow. It should be a simple process to to simplify some of them to bring the meaning into clear focus.
Author Response
Dear reviewers,
Hello! Thank you for your careful correction, I really learned, thank you! My modification instructions are as follows:
1.Your review comments“51-58 This is an example of a long sentence that is a bit hard follow for meaning. It could be turned into 3 sentences that might be easier to follow.
My answer is to simplify this sentence after it has been modified in the text
2.58-65 This sentence could be split at “… .strength and endurance. In these studies, the common combination of pressurized resistance training modes included … ..
My answer is to simplify this sentence after it has been modified in the text
3.159-165 This is another series of statements separated by semi-colons. You might consider separate sentences for easier reading.
My answer is to simplify this sentence after it has been modified in the text
4.216-221 Another example of a long sentence which could be divided.
My answer is to simplify this sentence after it has been modified in the text
5.223-226 This too is difficult to follow due to the stringing of multiple ideas in one sentence.
My answer is to simplify this sentence after it has been modified in the text
6.251 “The EMG data were statistically analyzed……… .."
My answer is to simplify this sentence after it has been modified in the text
7.282-286 Consider separating these three or four analyses into different sentences for easier reading.
My answer is to simplify this sentence after it has been modified in the text
8.310-316 Again, consider separating these semi-colon phrases into different sentences for easier reading. "For long sentences in the above position, Semicolons and difficult sentences have been modified.
My answer is to simplify this sentence after it has been modified in the text
9.“Should this be “In that study…” since you are referring to Cha’s basketball study?” .
My answer is: Yes, I mean the basketball article
10.“Does [7, 16-21] belong here since it appears you are referring the Cha’s work?”
My answer is Yes.
11.“Would it be easier to read this if you started a new sentence here with “In the present study, … ." ?"
My answer is: it is not easy to read, so in the revision I wrote, I changed the sentence structure to make it easier to read
12.“To what study does “in the above study” refer; is it Lei et al.? Should you give a citation number to clarify?”
My answer is: Referring to Lei et al., in the revision, I added the citation
13.“391 At what time does your phase “at this time” refer?”
My answer is: this part is indeed I did not write specific, I modify in the article, add a specific time.
14.“399 Should “Che” be Cha?”
and
“402 Should “Che” be Cha and should you provide a reference number for it?”
and
“481 Should “Che” be Cha?”
My answer: either "Cha" or "Che", I also corrected and added the reference.”
15.“405 Does the phrase “even better than that of the pectoralis major” belong here since you mentioned the pectoralis major before that? "
My answer is: Yes, it belongs here. The deltoids are a co-mover in the bench press, not the main power muscle group, but pressure training improves the activation of the deltoids significantly more than the active pectoralis major muscle
16.“402-409 Again, this is a very long sentence that is hard to follow."
My answer is that I have modified these sentences to be easier to understand.
17.410-417 This is another long sentence that is difficult to read and comprehend the meaning.
My answer is that I have modified these sentences to be easier to understand.
18.431-454 More long sentence's that would read better if they were divided. "
My answer is that I have modified these sentences to be easier to understand.
19.”438 Does “On the one hand” belong here since it was stated earlier?”
My answer is: Yes, it belongs here.
20.“453 What does “the big arm” mean?”
My answer is: the meaning of upper limb has been modified in the article.
21.“465 Does the reference [11] belong here?”
My answer is: Yes, it belongs here.
22."483"... .. having basketball players perform… ."
My answer is: I have revised this sentence to make it easier to understand
Thanks again for your guidance!
Best Wishes!
Reviewer 2 Report
Comments and Suggestions for Authors
DEAR AUTHORS
AFTER CHECK THE FIRST REVIEW MAJOR CHANGES ARE PROPOSED
KING REGARDS
Line 5-9. Include emails and complete affiliation of all authors
Line 12. Include p and data in the abstract section
Line 40-49. This first sentence is so long and divide in 3 sentences with differents references
Subjects: Correct trought the text, better participants
Line 97. 10 bodybuilders. Did you ckech the G-power
Line 97. In my opinión you have to include inclusion and exclusion criteria
Line 108. Ethical Issues. HELSINKI DECALRATION. Cite
World Medical Association Declaration of Helsinki: ethical principles for medical research involving human subjects.World Medical Association. JAMA. 2013 nov 27;310(20):2191-4. doi: 10.1001/jama.2013.281053.
Line 110. Weight. Better body mass
Line 112. Weight. Better body mass. Correct trought the text
Line 253. Repeated-measures two-way ANOVA. Did you checked LEVENE TEST BEFORE?
Line 328. Start the introducction part wiht main goal and main concusions
Line 351. “differences between the studies”. among studies
Line 354.” he 3rd and 4th”. r and th in superindex
Line 359.” The possible reasons can be attributed to the”. please include reference
Line 367. “Theoretically, the bodybuilders underwent more resistance training and may have had greater neuromuscular recruitment of the pecoralis major muscle during the bench press”. If this is a potential explanation, please include reference
Line 390. “Based on the quantitative relationship above, it can also be hypothesized that the metabolic pressure at this time may exceed the limit of the subjects’ recovery ability, suggesting that this may be related to the accumulation of greater fatigue due to the high occlusion pressure”,
Line 452. “may be because during the pressurized bench press training, part of the pressur- 452 ized band is tied to the proximal end of the big arm, which covers the triceps brachii, and 453 the occlusive and metabolic pressures on the triceps brachii have peaked, so the fatigue is 454 more substantial”. Agree but it needs a reference
Line 501, Include PRACTICAL APPLICATIONS, LIMITATIONS, STRENGTH AND FUTURE DIRECTIONS
Line 518. Rewrite references acoording MPDI RULES
Comments on the Quality of English Language
MODERATE CHANGES
Author Response
Dear reviewers,
Hello! Thank you for your careful correction, I really learned, thank you! Xia Ming is my modification description:
1.Reviewer: "Line 5-9. Include emails and complete affiliation of all authors"
My answer: Modification process has been added.
2.Reviewer: "Line 12. Include p and data in the abstract section"
My answer: Modification process has been added.
3.Reviewer: "Line 40-49. This first sentence is so long and divide in 3 sentences with differents references"
My answer is: It has been revised in accordance with the revised opinions.
4.Reviewer: "Line 97. 10 bodybuilders. Did you ckech the G-power"
My answer is: Yes, I calculated the sample size of the subjects with G-power.
5.Reviewer: "Line 97. In my opinion you have to include inclusion and exclusion criteria"
My answer is: it has been added in the revised draft.
6.Reviewer: "Line 108. Ethical Issues. HELSINKI DECALRATION. Cite"
My answer is: Yes, this study complies with the Declaration of Helsinki and has been supplemented in the text.
7.Reviewer: "Line 110. Weight. Better body mass
Line 112. Weight. Better body mass. Correct trought the text ”
My answer is: the text has been revised as suggested.
8.Reviewer: "Line 253. Repeated-measures two-way ANOVA. Did you checked LEVENE TEST BEFORE?"
My answer is: Yes, I conducted the LEVENE TEST prior to this, with Repeated-measures two-way ANOV on the basis of significant differences.
9.Reviewer: "Line 351." differences between the studies. among studies."
My answer is: this mutual language is not careful enough, has been revised in the article.
10.Reviewer: "Line 351." differences between the studies. "among studies
Line 354.” he 3rd and 4th”. r and th in superindex”
My answer is: the expression here is indeed not rigorous, I have revised in the article
11.Reviewer: “Line 367. “Theoretically, the bodybuilders underwent more resistance training and may have had greater neuromuscular recruitment of the pecoralis major muscle during the bench press”. If this is a potential explanation, please include reference
Line 390. “Based on the quantitative relationship above, it can also be hypothesized that the metabolic pressure at this time may exceed the limit of the subjects’ recovery ability, suggesting that this may be related to the accumulation of greater fatigue due to the high occlusion pressure”,
Line 452. “may be because during the pressurized bench press training, part of the pressur- 452 ized band is tied to the proximal end of the big arm, which covers the triceps brachii, and 453 the occlusive and metabolic pressures on the triceps brachii have peaked, so the fatigue is 454 more substantial”. Agree but it needs a reference”
My answer is: References have been added to the revised draft for support.
- Reviewer: "Line 501, including practical applications, limitations, advantages and future directions"
My answer: It has been supplemented in the revised draft
- Reviewer: "Line 518. Rewrite references to Harmonized MPDI RULES"
My answer is: All references have been modified to the format required by the journal.
Thanks again for the expert review.
I wish you good health and success in your work!
Best Wishes!
Round 2
Reviewer 2 Report
Comments and Suggestions for Authors
DEAR AUTHORS:
Minor changes are requiered
LINE 103-104. Better participants than subjects. Correct trought the text..
INCLUDE LEVENE TEST IN STATISTICAL PARAGRAPH
LINE 165. BODY MASS BETTER THAN WEIGHT
KING REGARDS
Comments on the Quality of English Language
MINOR
Author Response
Authors' Responses to Reviewer's Comments (Reviewer 2)
Dear reviewers,
Hello! Thank you for your careful correction, I really learned, thank you! Here are my notes on the modifications:
1.Reviewer: LINE 103-104. Better participants than subjects. Correct trought the text..
My answer: I've replaced participants with subjects.
2.Reviewer: INCLUDE LEVENE TEST IN STATISTICAL PARAGRAPH
My answer: I have added the LEVENE results to the text.
3.Reviewer:LINE 165. BODY MASS BETTER THAN WEIGHT
My answer is: I've replaced BODY MASS with WEIGHT.
Thanks again for the expert review.
I wish you good health and success in your work!
Best Wishes!
All authors